# Phytohormones and Transcriptome Analyses Revealed the Dynamics Involved in Spikelet Abortion and Inflorescence Development in Rice

**DOI:** 10.3390/ijms23147887

**Published:** 2022-07-17

**Authors:** Asif Ali, Tingkai Wu, Zhengjun Xu, Asad Riaz, Ahmad M. Alqudah, Muhammad Zafar Iqbal, Hongyu Zhang, Yongxiang Liao, Xiaoqiong Chen, Yutong Liu, Tahir Mujtaba, Hao Zhou, Wenming Wang, Peizhou Xu, Xianjun Wu

**Affiliations:** 1State Key Laboratory of Crop Gene Exploration and Utilization in Southwest China, Rice Research Institute, Sichuan Agricultural University, Chengdu 611130, China; asifali@sicau.edu.cn (A.A.); wtksicau@163.com (T.W.); mywildrice@aliyun.com (Z.X.); zhanghysd@163.com (H.Z.); liaoyongxiang123@163.com (Y.L.); xiaochenq777@126.com (X.C.); liuyutong910617@163.com (Y.L.); zhouhao666@foxmail.com (H.Z.); j316wenmingwang@163.com (W.W.); 2College of Agriculture and Biotechnology, Zhejiang University, Hangzhou 310058, China; asad.riaz76@gmail.com; 3Department of Agroecology, Aarhus University at Falkebjerg, Forsøgsvej 1, 4200 Slagelse, Denmark; ahqudah@gmail.com; 4Department of Grassland Science, College of Animal Science and Technology, Sichuan Agricultural University, Chengdu 611130, China; m.zafarsindhu@hotmail.com; 5Department of Biotechnology, School of Natural Sciences and Engineering, University of Verona, 37134 Verona, Italy; herrsoban@gmail.com

**Keywords:** phytohormone, spikelet abortion, transcriptome, DEG, inflorescence development

## Abstract

Panicle degeneration, sometimes known as abortion, causes heavy losses in grain yield. However, the mechanism of naturally occurring panicle abortion is still elusive. In a previous study, we characterized a mutant, *apical panicle abortion1331 (apa1331),* exhibiting abortion in apical spikelets starting from the 6 cm stage of panicle development. In this study, we have quantified the five phytohormones, gibberellins (GA), auxins (IAA), abscisic acid (ABA), cytokinins (CTK), and brassinosteroids (BR), in the lower, middle, and upper parts of *apa1331* and compared these with those exhibited in its wild type (WT). In *apa331,* the lower and middle parts of the panicle showed contrasting concentrations of all studied phytohormones, but highly significant changes in IAA and ABA, compared to the upper part of the panicle. A comparative transcriptome of *apa1331* and WT apical spikelets was performed to explore genes causing the physiological basis of spikelet abortion. The differential expression analysis revealed a significant downregulation and upregulation of 1587 and 978 genes, respectively. Hierarchical clustering of differentially expressed genes (DEGs) revealed the correlation of gene ontology (GO) terms associated with antioxidant activity, peroxidase activity, and oxidoreductase activity. KEGG pathway analysis using parametric gene set enrichment analysis (PGSEA) revealed the downregulation of the biological processes, including cell wall polysaccharides and fatty acids derivatives, in *apa1331* compared to its WT. Based on fold change (FC) value and high variation in expression during late inflorescence, early inflorescence, and antherdevelopment, we predicted a list of novel genes, which presumably can be the potential targets of inflorescence development. Our study not only provides novel insights into the role of the physiological dynamics involved in panicle abortion, but also highlights the potential targets involved in reproductive development.

## 1. Introduction

Rice (*Oryza sativa* L.) is the world’s 2nd most important cereal crop, providing food for one-half of the world’s population [1]. Its yield is dependent on three main agronomic traits, i.e., number of panicles, number of grains per panicle, and grain weight [2]. Therefore, the panicle and its bearing spikelets play a direct, key role in yield, and the achievement of an ideal panicle structure, size, and shape is a recent target of molecular breeders [3]. Panicle/spikelet abortion, alternatively called panicle/spikelet degeneration, occurs due to physiological defects and biochemical processes, which leads to a serious reduction (20–60%) in grain yield [4,5,6]. A few recent studies regarding the cloning of genes that control the whole panicle, or apical spikelets, have provided a coherent molecular understanding of panicle abortion [5,7]. Mutants showing the defects in spikelet development served as an important source to study the genetic basis of panicle abortion. Studies conducted on different mutants, e.g., *panicle apical abortion 1* (*paab1)* and *tutou1 (tut1)* were generated through tissue culture transformation and gamma rays, respectively [8,9]. A targeted mutagenesis tool, CRISPR/Cas9, was applied to edit the *DENSE AND ERECTILE PANICLE 1 (DEP1)* to increase yield by artificial mutagenesis [10]. Mutants produced through the CRISPR tool usually showed dysfunction in only one gene. However, mutants, e.g., *degenerated panicle and partial sterility 1 (dps1)* and *panicle apical abortion1019 (paa1019)* were produced through ethyl methanesulfonate (EMS) [6,11]. Previous studies revealed that EMS-induced mutants showed high frequencies of single nucleotide polymorphism (SNP s), which divulged that these mutants showed defects in multiple genes compared to the single gene defects induced by the CRISPR/cas9 system [12,13]. Hence, the comparative transcriptomic studies of EMS-induced mutants can reveal the novel genes and their molecular and biological mechanisms, including panicle development. A recent single-cell transcriptomic study profiled 37,571 inflorescence cells and provided insights into the auxiliary meristem to floret development in rice [14]. Wang et al. have reported the differential expression of 357 out of 22,000 genes in panicle development [15], which suggests that despite the discovery of abundant genetic factors controlling panicle development, many of these have not yet been cloned and functionally characterized.

Previous studies have revealed the molecular basis of spikelet abortion, e.g., increased reactive oxygen species (ROS), oxidative stress, abnormal programmed cell death (PCD), decrease in cell viability, and excessive DNA fragmentation [16,17]. These hyper-responses are generated in the cell due to the presence of an SNP, or a mutation in their targeted genes, proteins, and transcription factors [4,17]. ROS are produced during aerobic metabolism and consist of a singlet oxygen (^1^O_2_), hydroxyl radical (HO^.^), superoxide ion (O_2_), and hydrogen peroxide (H_2_O_2_) [18]. These molecules are produced in different cellular mechanisms, such as the electron transport chain, respiration, peroxidation, and the oxidation of different cellular metabolites [19]. However, we did not list molecular factors here in detail, as they were not the primary focus of this study. Apart from the molecular basis, different physiological factors caused by, e.g., drought, temperature stress, and environmental conditions, also influence spikelet development and can increase the rate of spikelet abortion [17,20]. Different abiotic stresses, e.g., water stress, as well as nitrogen and phosphorus (P) deficiencies, have also been reported to regulate the reproductive development of a floret [21,22,23]. Panicle abortion mostly occurs at the base or apical part of the panicle and usually occurs during the panicle elongation stage. The physiological occurrence of abortion in spikelets was only partially explained by both prevalent classical theories, e.g., resource limitation and self-organization. In the resource limitation theory, the supply of the essential nutrients required for the development and growth of inferior spikelets becomes insufficient, resulting in the abortion of the spikelets [24]. Transporters of aluminum and P, e.g., *aluminum-activated malate transporter (OsALMT7)* and *Pi transporter 1 (PHT1)* genes respectively, are typical examples of resource limitation [8,23]. According to the self-organization theory, the PCD in spikelets is induced as a result of endogenous signals [25]. Most of the studies reported so far in which degeneration was caused by increased PCD and ROS are in favor of self-organization. Despite multiple pieces of evidence in support of these theories, there are many questions left unanswered, e.g., why the spikelets of abortion have been mainly reported in the apical portions of all mutants, whether the naturally occurring abortion is relevant to the differential level of phytohormones, and whether apical spikelets have a different load of phytohormones under abortion. These are all questions that remain intriguing for researchers. The involvement of all factors of self-organization or resource limitation are regulated by different phytohormones [26,27], although some findings elucidating the individual role of phytohormones have highlighted their importance in spikelet development or panicle abortion. A previous study by Ali et al. has indicated the role of individual phytohormones in spikelet abortion [4]. However, whether and how their contents fluctuate in the different parts of inflorescence has not been previously reported. 

In a previous study, we presented the functional characterization of an EMS-generated mutant, *apical panicle abortion 1331 (apa1331),* which showed divergence from its wild type (WT) at the 6 cm stage of panicle development [5]. In this study, we quantified the five phytohormones in the upper, middle, and lower spikelets of an inflorescence and performed a comparative transcriptome analysis of upper spikelets to obtain further insight into the physiological basis of panicle abortion. Our study highlights the physiological basis of spikelet abortion and predicts the novel genes involved in panicle development.

## 2. Results

### 2.1. Apical Spikelets of apa1331 Showed a Differential Load of Phytohormones, Especially IAA and ABA

To discover whether the phenotype of spikelet abortion was associated with phytohormones, we quantified gibberellins (GA), auxins (IAA), abscisic acid (ABA), cytokinins (CTK), and brassinosteroids (BR) in *apa1331* from its divergence point and compared it with this same stage of WT spikelets (Figure 1). The stages of panicle development were determined by a previous study of Zhang et al. [28]. Previous studies indicate the dynamic changes in the phytohormones in different parts of the panicle under normal and stress conditions [27,29,30]. Therefore, we quantified the phytohormones from the lower, middle, and upper parts of the panicle separately. In *apa1331,* the quantification of GA revealed significantly (*p* < 0.01) decreased concentrations in the lower and middle panicles, but its level was significantly increased in the upper spikelet (Figure 1A). In *apa1331,* the concentrations of IAA were found to be significantly (*p* < 0.001) decreased in the lower and upper panicles compared to its WT (Figure 1B). Similarly, the concentration of ABA was also found to be significantly increased in the upper panicle of *apa1331* inflorescence (Figure 1C). In *apa1331,* concentrations of CTK were significantly decreased in the lower and middle, but increased in the upper spikelet, although concentrations of BR were also significantly decreased in all parts of *apa1331* compared to the WT spikelet (Figure 1D). These results revealed that IAA and ABA were found to be significantly decreased and increased, respectively, in the upper part of *apa1331* compared to its WT. However, the role of GA, CTK, and BR cannot be neglected. Therefore, the antagonistic phytohormone concentrations among inflorescence parts can help us to explain the mechanism underlying spikelet abortion.

### 2.2. Identification of DEGs in WT and apa1331

To investigate the gene expression associated with the apical spikelet abortion in *apa1331*, in addition to changes in phytohormones, we isolated the RNA (three biological repeats) from the apical spikelets of *apa1331* and WT at the 6 cm inflorescence tissues and constructed cDNA libraries for sequencing. A range of 16.97–18.69 million single-end 50-base reads was produced from each library (Figure 2A). Using a minimal count per million (CPM) 0.5 reads, counts were transformed for clustering and principal component analysis (PCA). Using strict cut-offs for gene expression, 32,582 genes were uniquely mapped, among which 21,724 genes passed the filters (Appendix A). PCA analysis of gene expression revealed 52% and 24% variance among *apa1331* vs. WT samples, respectively (Figure 2B). PCA analysis showed the presence of significant variations between *apa1331* and WT, which probably played a role in causing spikelet abortion. Using the DESeq2 comparison of all groups, 1587 and 978 genes were found down and upregulated in *apa1331,* at a minimum fold change (FC) > 2, and an adjusted false discovery rate (FDR) < 0.1 (Figure 2C). A heat map of significantly down and upregulated DEGs showed enrichment of different pathways in *apa1331* compared to WT (Figure 2D). The expression profile of variations in DEGs was determined by clustering analysis based on the k-means method (calculated using the Pearson correlation), and significantly enriched DEGs in *apa1331* were divided into four clusters (Figure 2E). K-means enrichment analysis revealed 966, 262, 348, and 424 genes in clusters A, B, C, and D, respectively. The top three relevant genes found in cluster A were related to cell wall biogenesis, the H_2_O_2_ catabolic process, and pollen development. Most significantly, enrichment genes were related to DNA replication and negative regulation of peptidase activity in clusters B and D; however, cluster C did not show any specific consensus algorithm. Together, these data revealed that the phenotype of apical spikelet abortion was associated with significant variations in dozens of genes, revealed through differential expression and k-means clustering.

### 2.3. Hierarchical Clustering and GO Terms Analysis of DEGs 

As the spikelet abortion was associated with a massive response that resulted in significant down- and upregulated DEGs, hierarchical clustering was performed to visualize the functional analysis of GO terms and their respective relationships based on the distance among overlapped genes [31,32]. GO terms associated with DEGs were downregulated in biological processes related to plant secondary cell wall biosynthesis, plant cell wall biogenesis, and photosynthesis (Figure 3A). Perhaps this is the reason why the downregulation of the components of the cell wall causes a softening of the tissues, and a decrease in wax and cutin contents causes an excessive loss of water, which ultimately causes abortion in the apical spikelets [5,33]. The significantly upregulated biological processes associated with DEGs were related to cell division and DNA conformational changes. The significant upregulation of DNA conformational changes can be associated with excessive PCD and DNA fragmentation in *apa1331* [5,34].

Significantly enriched cellular components associated with these DEGs were mainly downregulated in terms of extracellular components, apoplast, thylakoid, and photosystem and upregulated in terms of protein DNA complexes, non-membrane-bounded organelles, and microtubules (Figure 3B). Significantly enriched molecular functions associated with these DEGs were downregulated in terms of antioxidant activity and peroxidase activity and upregulated in terms of protein heterodimerization activity, serine endopeptidase inhibitor activity, and DNA binding (Figure 3C). The interaction network of GO terms associated with DEGs was majorly related to the downregulation of peroxidase activity, antioxidant activity, and oxidoreductase activity and the upregulation of the tetrapyrrole groups. 

### 2.4. KEGG Pathway Expression Analysis of DEGs 

The KEGG expression of DEGs across all the samples of WT and *apa1331* were identified using PGSEA at FDR cutoff < 0.1 and *p* < 0.05 [35]. PGSEA can be used to identify the list of genes that were downregulated in a specific dataset.. It displays the FC activities of genes compared to their mean, in terms of z-score, in a specific pathway. Pathway analysis revealed that the gene-regulating cell wall macromolecules biosynthetic processes, cell wall polysaccharides biosynthetic processes, and plant-type cell wall biogenesis were downregulated in *apa1331.* These results are consistent with the enrichment analysis (Figure 4A). In addition, the DEGs are related to fatty acid derivative metabolic processes and other organic metabolites. Notably, using the criteria mentioned above, PGSEA did not yield a set of genes related to any specific cellular components. Pathway analysis of molecular functions revealed the downregulation of amine-lyase activity, cullin family protein binding, and exonuclease activity, active with either ribo or deoxyribonucleic acid (Figure 4B). Meanwhile, terpene synthase activity, carbon-oxygen lyase activity, phosphate activity, β-glucose activity, and magnesium ion binding were found to be upregulated in *apa1331* compared to WT.

### 2.5. Transcriptome Response Revealed the Key Genes Involved in Panicle Development

As the above-mentioned DEGs revealed a significant response of transcriptome in aborted spikelets of *apa1331* compared to its WT, we further analyzed the fold enrichment analysis and retrieved expression data from the MSU expression network profile of the rice genome annotation project [36,37]. Based on the GO enrichment analysis of DEGs and their significant expression and variation spectrum during late inflorescence, early inflorescence, and anthers, we predicted the genes that might regulate the phenotype of spikelet abortion in *apa1331* (Figure 5, Table 1). In prediction analysis, among the 16 downregulated DEGs, only two, *OsC6 (LOC_Os11g37280)* and *OsBiP4 (LOC_Os05g35400),* have already been cloned and reported to regulate the panicle development and endoplasmic reticulum stress in rice, respectively [38,39].

### 2.6. Quantitative Real-Time Polymerase Chain Reaction (RT-qPCR) Validation of Genes Expression 

To validate the integrity of the RNA-seq expression data, we randomly selected four genes from up- and downregulated DEGs in *apa1331* for RT-qPCR analysis. The comparison of the FC values obtained from the transcriptome and RT-qPCR data were consistent with that of the RNA-seq data (Appendix A). 

## 3. Discussion

Inflorescence development is an important and complex biological process that is directly influenced by endogenous cues. Consequently, the development of spikelet organs (including spikelet development) is controlled by several factors, and changes lead to abnormality or abortion in spikelets that in turn, affect the final grain yield. Therefore, understanding the role of endogenous factors, including phytohormones, on inflorescence and spikelet development is imperative to improve grain yield.

### 3.1. Role of Phytohormones in ROS Homeostasis and Spikelet Abortion 

Phytohormones are required in plants for different physiological functions. Apical spikelet abortion refers to the degeneration of cells at the growing tip, which produces auxin to inhibit the growth of auxiliary buds [42]. Whether the mechanism of apical dominance is affected by abortion or degeneration is still an unanswered question. The quantification of auxin (IAA) revealed its significantly decreased levels in the apical parts of the inflorescence of *apa1331* as compared to WT. Under normal conditions, apical dominance is achieved through the downward movement of auxin within the stem [42]. As the auxin cannot enter the buds, it activates the upward movement of CTK that promotes bud growth. The level of CTK in the apical parts of the inflorescence was higher in *apa1331* compared to WT. The level of CTK, essential for apical dominance, was higher in the upper parts of spikelets of *apa1331* compared to that of WT, which indicates the role of additional metabolites in causing abortion. Sugar molecules, which act as a signal for the growing tip, are also in strong demand for normal apical spikelet development. The deficiency of sugar molecules has also been reported to play a role in the balance of ROS and PCD in apical spikelet development [8], which may also be a cause of spikelet abortion in our study. A decrease in sugar consumption in the young panicle caused a significant reduction in spikelet number [43]. Sugar plays a role in combination with antioxidant enzymes, salicylic acid, and IAA to prevent spikelet degeneration [44]. Our findings suggested that phytohormones and sugars are crucial determinants of inflorescence development and spikelet abortion.

A strong genetic association of genes involved in phytohormones such as ABA, underlying the natural variation of spikelet abortion in the upper part of the spike, was found in barley [45]. Previous studies revealed that an increased level of ABA in the apical spikelet can enhance the risk of degeneration and decrease the rate of grain filling [46]. The previous study indicated the role of ABA in ROS signaling [46,47], which may influence other biological mechanisms of reproductive development. A higher level of ABA in *apa1331* apical parts of spikelets may be credited for increased ROS production and PCD. ABA has also been reported to interact with ethylene, and its exogenous application causes male sterility [48]. These findings highlight the essential role of ABA-mediated ROS production in apical spikelet abortion; however, further research should focus on clarifying the role of individual phytohormones. In addition, the question of why the lower and middle parts of inflorescence also contain a decreased contents of IAA in *apa1331* compared to WT should be investigated in the future, as apical dominance demands the movement of auxins in the middle and lower parts. Moreover, whether the contrasting concentrations of GA, ABA, CTK, and BR in the upper, middle, and lower parts are required for normal infloresence development, or prevail due to the phenotype of aborton should be investigated in detail in the future.

### 3.2. The Transcriptomic Response of DEGs Highlighted the Role of PCD and Increased ROS in Panicle Abortion

Recent applications of high throughput sequencing, coupled with computational genomics tools, have attracted the focus of researchers to study large-scale transcriptome profiling. A previous comprehensive transcriptome profiling study has revealed the dynamics involved in meiosis and male gametophyte development [49]. It has also been used to identify the physiological dynamics and response of anthers under cold, heat, and drought stress [49,50,51]. However, the detailed transcriptomic response under the stress of spikelet abortion has not been reported to date, despite the fact that several mutants and their candidate genes have been cloned. It is also important to know the DEGs, which are potentially involved in such phenomena, that empower us to understand how to improve the final grain yield. Spikelet abortion is mostly accompanied by innate plant responses that are produced due to the dysfunction of specific proteins. Previous studies have revealed that these proteins are involved in mitigating the deleterious effects produced in the endoplasmic reticulum and mitochondria [7,11,39]. The transcriptomic response of *apa1331* DEGs revealed that GO terms were associated with the downregulation of peroxidase activity, antioxidant activity, oxidoreductase activity, and the upregulation of tetrapyrrole groups. A decrease in the oxidoreductase and antioxidants activity and an increase in the ROS can cause damage to the cellular machinery [52]. The destruction of cells also prevails due to hyperactivation of the plant immune system and stimulation of caspases and proteases [53]. ROS are produced in the cellular environment during the signaling of photosynthesis, pathogen recognition, and stress perception, and are also produced to cope with the elevated levels of ROS [18]. Consequently, controlled PCD helps the plants to activate the different pathways, e.g., water-water cycle and ascorbate glutathione cycle, for scavenging ROS. These pathways release different antioxidants and enzymes to detoxify excessive ROS [54]. The detoxification of excessive ROS helps different organelles, such as the stroma and thylakoid membrane of the chloroplast, mitochondria, and peroxisome, by protecting them from photooxidative damage. A previous study reported that if the excessive ROS could not be detoxified by plants and PCD, it may cause an excessive DNA fragmentation and a decrease in cell viability, also causing the abortion of the spikelet due to hyper-response of inositol-requiring enzyme 1 (IRE1), which causes stress in the endoplasmic reticulum [55]. These findings support that spikelet abortion in *apa1331* is associated with increased ROS and hyper-immune response.

### 3.3. Role of Antioxidants and Redox Changes in Inflorescence Development and Spikelet Abortion 

Plants require different oxidants, ROS, and reactive nitrogen species (RNS) for many growth and development-related processes. Plants maintain the concentration of oxidants by releasing antioxidants in different pathways, as the elevated level of oxidants is also toxic to cellular function, including pollen development [56]. The reproductive development, especially microsporogenesis and pollen development, are sensitive to the external environment. A complex network of ROS and RNS controlling genes regulates the redox homeostasis by inducing PCD in the tapetum for normal inflorescence and pollen development [56]. Balance in the ROS production and scavenging in a cell is maintained by enzymatic (superoxide dismutase, peroxidase, catalase, etc.) and non-enzymatic antioxidants (carotenoids, flavonoids, etc.). However, if the level of ROS and RNS increase above or below a certain threshold, it eventually perturbs the redox homeostasis, resulting in premature pollen, failure in the microspore development, spikelet abortion, and complete sterility. The reason for partial or complete sterility has been reported to be caused by the deficient or improper supply of nutrients from delayed or premature tapetum degeneration [57]. Hierarchical clustering revealed the significant downregulation of antioxidant, oxidoreductase, and peroxidase activities, implying the persistence of an imbalanced redox environment in the spikelets of *apa1331*. Previous studies have also reported that a decrease in antioxidant activity and an imbalance in redox reactions causes panicle degeneration [7,56], and the tetrapyrrole group has also been reported to be involved in ROS homeostasis, PCD, and photosynthate assimilation [58]. These studies suggest that spikelet abortion in *apa1331* was caused due to imbalance in the redox homeostasis. 

### 3.4. Role of Non-Enzymatic Metabolites in Inflorescence Development and Spikelet Abortion 

In addition to enzymatic oxidants, non-enzymatic oxidants and transition elements have also been reported to play a role in fertility and inflorescence development. A recent study by Huang et al. has indicated the role of iron (Fe) in the fertility of anthers [59]. Similarly, P is an essential component of amino acid and carbohydrate metabolisms, and its deficiency is associated with decreased fertility and abortion due to photooxidative damage to the cellular organelles [60]. Glutathione (GSH), ascorbate (ASA), proline, tocopherol, phenolic acid, carotenoids, and flavonoids are non-enzymatic antioxidants which are suggested to play a role in pollen fertility [56,61]. GSH and ASA regulate the redox status during pollen development [61]. The deficiency of proline contents in Arabidopsis has been reported to cause abortion and abnormalities in reproductive development [62]. Significant upregulation of terpene synthase activity, carbon-oxygen lyase activity acting on phosphates, and magnesium ion binding indicate that their transport to the sink or supply from the source has been increased in *apa1331*, which leads to an imbalance in redox changes. High temperature has been reported to induce irreversible changes in plant cell wall invertase and cause sterility due to reduced accumulation of sucrose in microspores [63]. The downregulation of genes involving plant cell wall biogenesis in *apa1331* implies structural changes in cell wall due to excessive ROS and redox imbalance. KEGG enrichment analysis also suggested that changes in non-enzymatic metabolites and metallic ion-related genes can also be a potential source of promoting abortion and damage to apical spikelets in *apa1331*. Studies focussing on up- and downregulated DEGs in *apa1331*, especially whose functions are related to the cell wall, cytoskeleton, plastid, cytosol and endoplasmic reticulum, will open up new findings for understanding infloresence development and spikelet abortion.

## 4. Materials and Methods

### 4.1. Experimental Material

The panicle degenerated/abortion mutant *apa1331* was derived from an indica maintainer line Yixiang 1B (WT) by EMS mutagenesis. Additional details of its characterization, phenotype, and breeding have been reported in a previous study by Ali et al. [5]. The inflorescence development of *apa1331* was normal up to the 5 cm stage of panicle development, (the same as its WT); however, it started to show the abortion of spikelets at 6 cm (divergence point between the mutant and its WT). Three technical replicates of samples for RNA-seq data analysis were collected individually at their divergence point from the apical tissues of *apa1331* (degenerated) and WT (normal). Plants were grown under natural conditions in the experimental fields at the Rice Research Institute, Sichuan Agricultural University, Chengdu, China (N 30.67°, E 104.06).

### 4.2. Quantification of Endogenous Phytohormones

The concentrations of phytohormones were determined using enzyme-linked immunosorbent assay (ELISA) kits by following manufacturer protocols. Data were recorded from three technical repeats and represent the mean ± standard deviation (SD).

#### 4.2.1. Samples Preparation

A total of 800 mg of fresh tissues of *apa1331* and WT were used for the analysis of GA, according to the methods used in the previous study by Jahan et al. [64]. The tissues were blended with an extraction solution of 80% methanol (*v/v*) and 1 mM butylated hydroxytoluene. The extraction solution was incubated at 4 °C and centrifuged at 3500 rpm, and the supernatant was obtained. After washing, the supernatant was run through a cartridge and the residue was dissolved in 0.1% phosphate buffer solution (PBS) solution.

#### 4.2.2. Quantification of GA

The final concentration of GA was quantified according to a previous study by Zhou et al. [65] using an ELISA detection kit, product number: LE-Y1587, Lyle Biotechnology Co., Ltd., Nantong, China, according to the manufacturer’s instructions. Measurement of the absorbance (OD value) in each well was performed at the wavelength of 450 nm. 

#### 4.2.3. Quantification of IAA and ABA

The quantifications of IAA and ABA were evaluated by ELISA Phyto-IAA kit (96T) and ELISA-ABA kit (96T), respectively, purchased from Beijing Yonghui Biotechnology Co., Ltd., Beijing, China, according to the antibody–antibody–enzyme labeled complex. The addition of samples containing IAA/ABA and IAA/ABA-antibody labeled with HRP (horseradish peroxidase) gives a complex of antibody–antigen–antibody, which was thoroughly washed with substrate (3,3′,5,5′-Tetramethylbenzidine) TMB color. TMB is converted to blue color after the catalysis of the HRP enzyme and converted to yellow color with the conjugation of IAA/ABA. The shade of the color is correlated with the quantity of auxin, which was measured with a microplate reader at the wavelength of 450 nm. The concentration of IAA/ABA was calculated using the standard curve.

#### 4.2.4. Quantification of CTK and BR

Samples for CTK and BR were prepared according to Zhou et al. [65]. The final concentrations of CTK and BR were quantified using ELISA detection kits, product numbers: SP29778 and SP29798, respectively, Wuhan Saipei Biotechnology Co., Ltd., Wuhan, China, according to the manufacturer’s instructions. Measurement of the absorbance (OD value) in each well was performed at the wavelength of 450 nm. 

### 4.3. RNA Extraction and Illumina Sequencing

Total RNA was extracted using a Trizol reagent kit (Invitrogen, Carlsbad, CA, USA) according to the manufacturer’s protocol. RNA quality was assessed by Agilent 2100 Bioanalyzer (Agilent Technologies, Palo Alto, CA, USA) with the sample RNA integrity number RIN >7. The enriched mRNA was fragmented into short fragments using fragmentation buffer and reverse transcribed into cDNA with random primers. Second-strand cDNA was synthesized by DNA polymerase I, RNase H, dNTP, and buffer. The cDNA fragments were purified with a QIAquick PCR extraction kit (Qiagen, Venlo, The Netherlands) and the ends were added with poly (A), then ligated to Illumina sequencing adapters. The ligation products were size-selected by agarose gel electrophoresis and PCR amplified and sequenced using Illumina HiSeq TM2500 from Gene Denovo Biotechnology Co. (Guangzhou, China).

### 4.4. Reads Mapping and Annotations

Quality reads of the raw RNA-Seq data were processed by FASTP (version 0.18.0) according to the instructions of Chen et al. [66], and the short reads alignment tool Bowtie2 (version 2.2.8) was used in the sensitive local mode for the mapping of reads. The clean reads were further used for assembly. The index of the reference genome was established according to the genomic data of the Ensemble database [67]. The single-end clean reads were mapped to the reference genome of the *Oryza sativa* indica group with default parameters in iDEP. 0951 [31]. The raw data files used for transcriptomic analysis have been uploaded to the National Center for Biotechnology Information (NCBI) under the Bio Project PRJNA847685, biosamples (SAMN28950847, SAMN28950856, SAMN28950881, SAMN28951021, SAMN28951022, and SAMN28951023) and sequence read archive (SUB11685885). The datasets supporting the conculsions have been provided as a Appendix A.

### 4.5. Differential Expression Genes (DEGs) Identification

StringTie v1.3.1 was employed to count the number of reads mapped to each gene, and gene expression was quantified [68]. DESeq2 was employed to estimate the FC and differentially expressed genes (DEGs) from the read counts data of gene expression level, given in fragments per kilobase of exon per million mapped fragments (FPKM) [69]. The *p*-values were adjusted for multiple testing using the default method, integrated with iDEP 0.951 [31]. The genes/transcripts of DEGs were determined with the parameters using FDR below 0.1, adjusted *p* < 0.05, and absolute FC > 2.

### 4.6. Gene Function Annotation

Gene ontology (GO) enrichment analysis was performed using the functions of the hypergeometric distribution test for the calculation of GO terms described by Zheng and Wang et al. [70]. All DEGs were mapped to GO terms in the Gene Ontology database (www.geneontology.org, accessed on 20 May 2022) and the number of genes was calculated for each term, and the significantly enriched (FDR below 0.1) GO terms in DEGs compared to the genome background were defined by a hypergeometric test. The Kyoto Encyclopedia of Genes and Genomes (KEGG) was used for pathway analysis to retrieve the enriched pathway, using FDR < 0.1 as a threshold for significantly enriched DEGs. 

### 4.7. RT-qPCR Analysis

Apical spikelet tissues measuring 6 cm were used for RT-qPCR analysis. Ubiquitin (forward primer: AACCAGCTGAGGCCCAAGA, reverse primer: ACGATTGATTTAACCAGTCCATGA) showed the highest stability in its expression among all tissues and was used as an internal control. All the primers used for RT-qPCR analysis are listed in Appendix A. RNA extraction and PCR amplifications were performed according to the previous studies by Wu et al. [71].

## 5. Conclusions

Panicle degeneration causes a significant decrease in yield. To cope with this defect, an understanding of the physiological mechanism, in addition to its molecular control, is essential. Comparative analysis of WT and *apa1331* revealed that phytohormones tend to change in different parts of the panicle, due to either coping, or as a result of panicle abortion. Studies involving the individual role of phytochromes, related to apical dominance (GA and CTK) and abiotic stress (IAA and ABA), are needed to understand their contrasting concentrations in different parts of inflorescence. The DEGs, controlling antioxidant, peroxidase, and oxidoreductase activities, were the potential reason for the imbalanced redox status in *apa1331*. These DEGs tend to downregulate, causing an increased level of ROS and DNA fragmentation. Our study further strengthens the idea that the PCD and ROS have the potential to regulate spikelet development and its abortion, especially on the upper part of the inflorescence. In addition, we predicted novel targets to improve final grain yield by controlling spikelet abortion. These findings can be used in molecular and genetic validation for understanding the mechanism of grain yield improvement. Studies involving, e.g., RNA silencing, knockout, and overexpression of these predicted genes will be helpful to further validate their function in panicle development. 

## Figures and Tables

**Figure 1 ijms-23-07887-f001:**
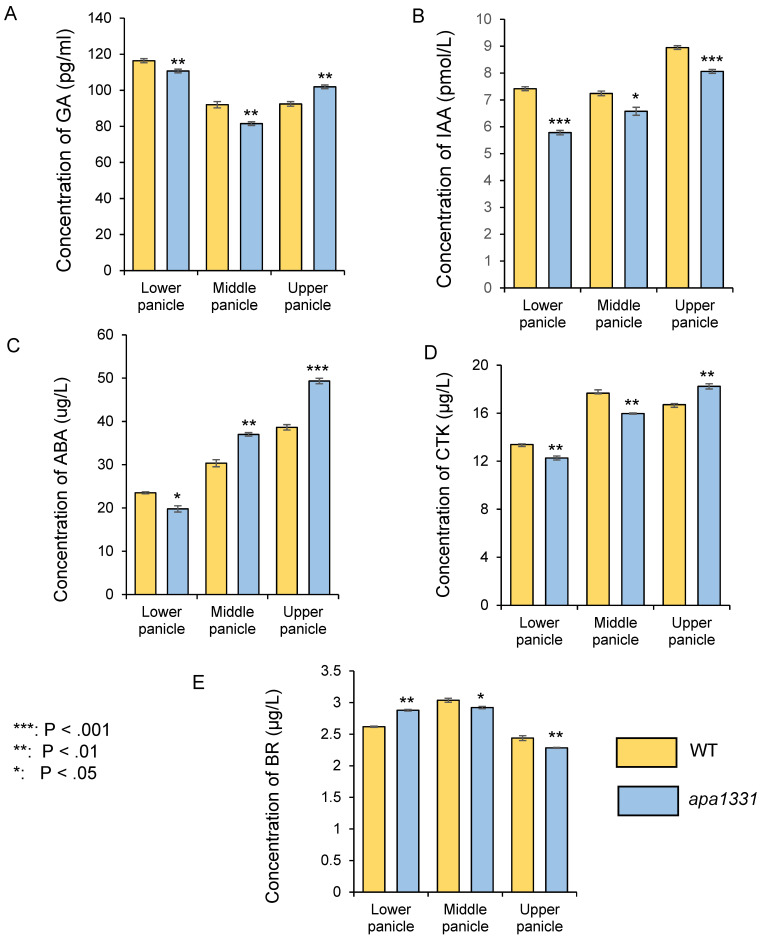
Comparative analysis of phytohormones in different parts (lower, middle, and upper) of the 6 cm panicle wild type (yellow) and *apa1331* (blue); gibberellic acid (**A**); indole 3 acetic acid (**B**); abscisic acid (**C**); cytokinin (**D**); and brassinosteroids (**E**). A Student’s *t*-test was used to analyze the significance of data, presented as mean ± standard deviation, where asterisks *, **, and *** show that *p* < 0.05, *p* < 0.01, and *p* < 0.001, respectively.

**Figure 2 ijms-23-07887-f002:**
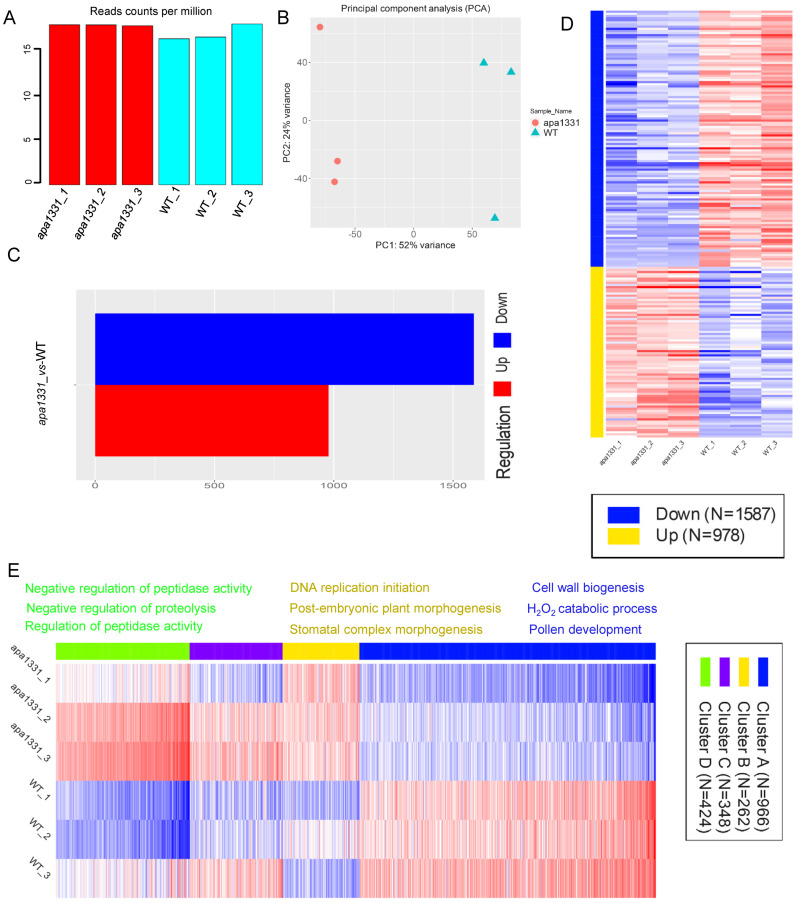
Identification of DEGs in WT and *apa1331.* Read counts per million (**A**); principal component analysis (**B**); difference between the number of up- and downregulated genes (**C**); heat map displaying the clustering analysis of DEGs in up and downregulation (**D**); heat map displaying the expression of DEGs based on k-means enrichment analysis (**E**).

**Figure 3 ijms-23-07887-f003:**
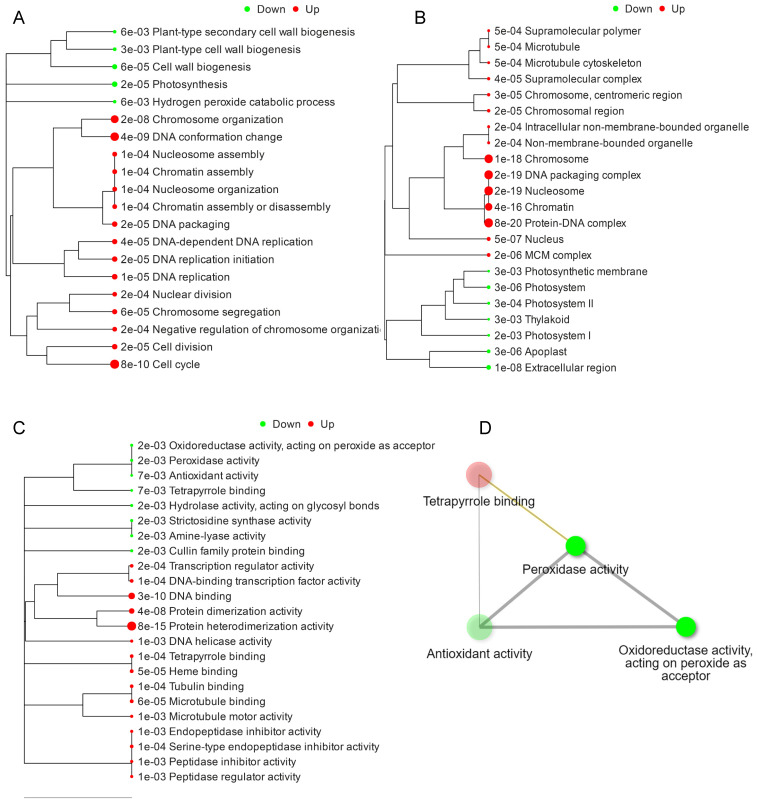
Hierarchical clustering based on GO functional analysis. GO biological processes (**A**); GO cellular processes (**B**); GO molecular functions associated with DEGs (**C**); and GO terms network interaction analysis of DEGs (**D**). DEGs were subjected to design hierarchical clustering trees using iDEP 0.951, with parameters: most variable genes to include—2000, number of clusters—4, and normalize by gene—mean center.

**Figure 4 ijms-23-07887-f004:**
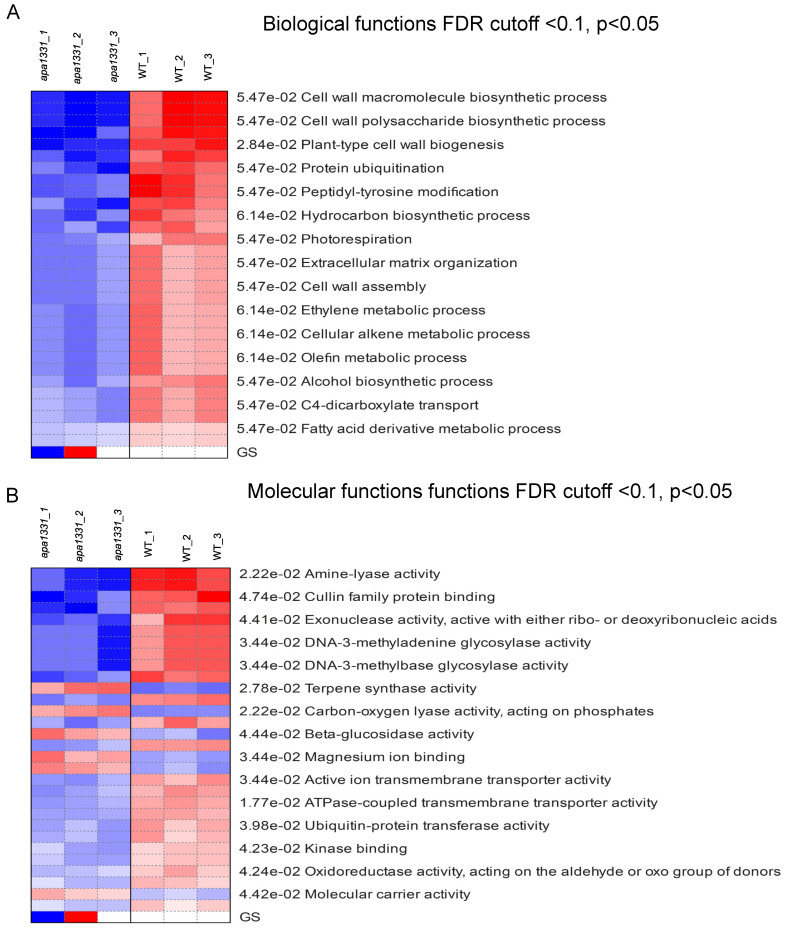
KEGG expression profiles of DEGs using PGSEA. KEGG terms associated with biological functions of DEGs using PGSEA (**A**); KEGG terms associated with molecular functions of DEGs using PGSEA (**B**). Red and blue indicate upregulated and downregulated genes, respectively.

**Figure 5 ijms-23-07887-f005:**
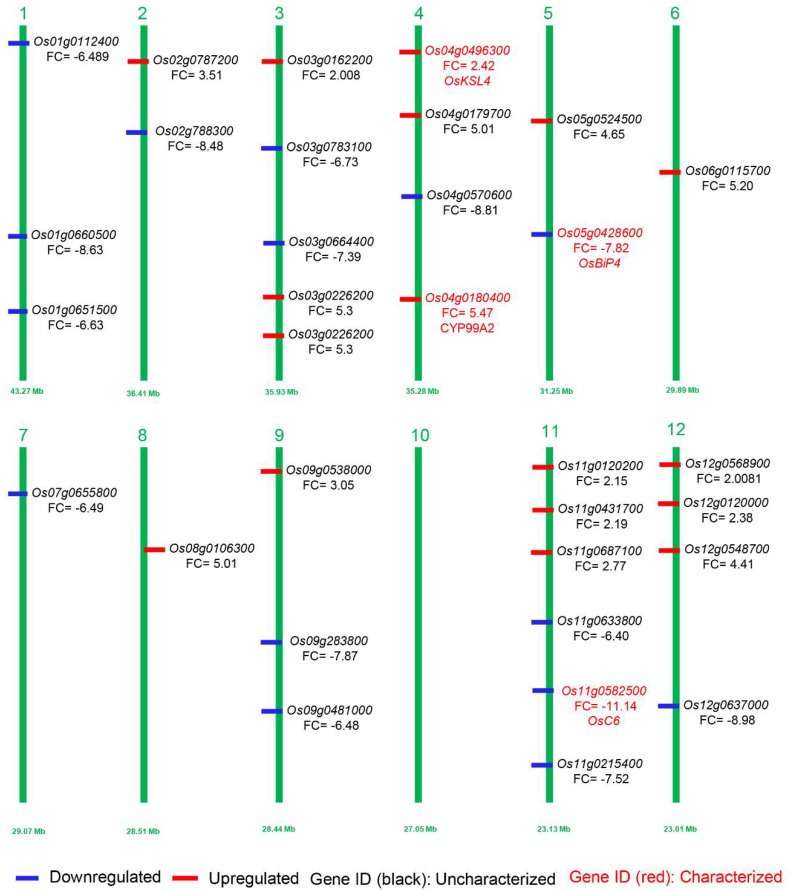
Visualization of predicted genes possibly involved in inflorescence development and panicle abortion. Downregulated and upregulated DEGs in *apa1331* are mentioned in blue and red colors, respectively. Perviously cloned and uncloned genes are mentioned in red and black text, respectively. The numbers 1–12 represent the number of chromosomes of rice.

**Table 1 ijms-23-07887-t001:** List of downregulated DEGs, GO, and their expression spectrum in inflorescence and anthers.

RAPD	MSU	FC	P- Adjusted	Late Inflorescences (FPKM)	Early Inflorescence (FPKM)	Anther (FPKM)	GO
*Os11g0582500*	*LOC_Os11g37280*	−11.148	5.58 × 10^−14^	273.82	7.2958	19.027	lipid binding, transport, plasma membrane
*Os12g0637000 *	*LOC_Os12g44010 *	−8.9894	3.73 × 10^−13^	8.3933	0.8633	4.3885	cell wall, hyrdorlase activity, cytosol
*Os04g0570600 *	*LOC_Os04g48210 *	−8.8109	1.00 × 10^−11^	0.4118	6.4236	4.8603	catalytic activity, metabolic processes, oxygen binding
*Os01g0660500 *	*LOC_Os01g47100 *	−8.6318	6.12 × 10^−16^	0	0.7213	5.8671	nucleotide process, nuclease, nucleic acid binding
*Os02g0788300 *	*LOC_Os02g54690 *	−8.4887	1.42 × 10^−14^	0	0.6296	0.6264	translation, nucelic acid binding, cytoplasm
*Os12g0242700 *	*LOC_Os12g13930 *	−8.4829	4.58 × 10^−19^	0.4769	15.97	2.6067	plastid, transferase, lipidic metabolic process
*Os09g0273800 *	*LOC_Os09g10200 *	−7.8273	1.50 × 10^−10^	0	0.8435	0	cell cycle, cytoskeleton, nucleus
*Os05g0428600 *	*LOC_Os05g35400 *	−7.5769	1.77 × 10^−14^	0.2241	1.7673	0.5022	endoplasmic reticulum, response to biotic stress, cellular component
*Os11g0215400 *	*LOC_Os11g10910 *	−7.5261	7.58 × 10^−29^	0	10.882	0	protein metabolic process, hydrolase activity, chloroplast
*Os03g0664400 *	*LOC_Os03g46150 *	−7.3955	4.39 × 10^−5^	0	7.2273	0	lipid binding, transport, membrane
*Os01g0112400 *	*LOC_Os01g02190 *	−6.8985	0.000386131	0	47.986	0	transport, transporter activity, cellular process
*Os03g0783100 *	*LOC_Os03g56974 *	−6.7322	8.00 × 10^−8^	0	3.2233	0	topoisomerase function, nuclear antigen
*Os01g0651500 *	*LOC_Os01g46270 *	−6.6382	3.98 × 10^−7^	13.876	0	0	lipidic process, transferase activity, protein binding
*Os07g0655800 *	*LOC_Os07g46210 *	−6.4993	1.56 × 10^−72^	0	63.094	0.1016	ATPase activity, LTP family protein, protease inhibition
*Os09g0481000 *	*LOC_Os09g30320 *	−6.4816	6.41 × 10^−6^	704.44	62.081	178.28	BURP domain-containing protein
*Os11g0633800 *	*LOC_Os11g41560 *	−6.4084	1.52 × 10^−5^	3.31	1.9748	4.2554	F-box domain-containing protein

Based on fold enrichment analysis and GO expression of upregulated DEGs in late inflorescence, early inflorescence, and anthers, we predicted 15 genes; among them, only 2 genes, i.e., *KAURENE SYNTHASE-LIKE 4* (*OsKSL4)* and CYTOCHROME P450 MONO-OXYGENASES CYP99A2, have been cloned so far and found to be involved in GA metabolism and diterpenoid phylotoxins [40,41]. The above prediction analysis reveals the novel genes that might be involved in the panicle development by regulating the expression of early or late inflorescence or anther development, directly or indirectly (Figure 5, Table 2).

**Table 2 ijms-23-07887-t002:** List of upregulated DEGs, GO, and their expression spectrum in inflorescence and anthers.

RAPD	MSU	FC	P-adjusted	Late Inflorescences (FPKM)	Early Inflorescence (FPKM)	Anther (FPKM)	GO
*Os03g0162200*	*LOC_Os03g06670*	2.008056	2.68 × 10^−5^	10.1154	108.937	2.40101	flower development, DNA binding, response to external stimulus
*Os12g0568900*	*LOC_Os12g38120*	2.089498	1.86 × 10^−9^	2.32593	66.3759	0	response to biotic stimulus, membrane
*Os11g0120200*	*LOC_Os11g02740*	2.15798	5.63 × 10^−5^	1.48597	6.96247	0.79251	cytosol, biological processes
*Os11g0431700*	*LOC_Os11g24374*	2.299897	3.29 × 10^−10^	0.637085	1.49044	0	cell, protein metabolic processes, hydrolase activity
*Os12g0120000*	*LOC_Os12g02710*	2.383414	5.15 × 10^−8^	1.29036	6.28534	0.768418	molecular functions, cytosol, biological process
*Os04g0496300*	*LOC_Os04g41900*	2.429537	3.13 × 10^−9^	10.2279	114.047	3.76954	molecular functions, cellular components, biological process
*Os11g0687100*	*LOC_Os11g45990*	2.770047	0.000458	3.23511	22.0661	0	protein modification, binding, catalytic activity
*Os09g0538000*	*LOC_Os09g36700*	3.053863	2.19 × 10^−22^	90.9664	36.714	0.399433	cell, nuclease activity, protein modification
*Os02g0787200*	*LOC_Os02g54590*	3.510367	4.94 × 10^−5^	0.516972	0.439793	94.1342	metabolic processes, cellular processes, response to stress
*Os12g0548700*	*LOC_Os12g36240*	4.411432	7.2 × 10^−5^	0	0	0.835161	response to stress, metabolic stress, cell wall
*Os04g0179700*	*LOC_Os04g10060*	5.012307	6.83 × 10^−5^	0.498632	0	0	catalytic activity, plastid, lipid metabolic processes.
*Os08g0106300*	*LOC_Os08g01520*	5.014211	4.17 × 10^−9^	0	0.479214	0	binding, catalytic activity, oxygen binding
*Os06g0115700*	*LOC_Os06g02530*	5.207817	6.28 × 10^−6^	8.49399	37.2876	1.03739	molecular function, biological process
*Os03g0226200*	*LOC_Os03g12510*	5.306009	3.32 × 10^−6^	0.871437	0	2.50797	cell wall, cytosol, oxygen binding
*Os04g0180400*	*LOC_Os04g10160*	5.479893	4.09 × 10^−5^	0.492373	0	0	metabolic process, cytosol, oxygen binding

## Data Availability

The raw data files used for transcriptomic analysis have been uploaded to the National Center for Biotechnology Information (NCBI) under the Bio Project PRJNA847685, biosamples (SAMN28950847, SAMN28950856, SAMN28950881, SAMN28951021, SAMN28951022, and SAMN28951023), and sequence read archive (SUB11685885).

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
