# Peer review of "Phytohormones and Transcriptome Analyses Revealed the Dynamics Involved in Spikelet Abortion and Inflorescence Development in Rice"

_ijms, 2022, doi:10.3390/ijms23147887_

Round 1

Reviewer 1 Report

Authors submitted a manuscript entitled “Phytohormones and Transcriptome revealed the dynamics involved in spikelet abortion and inflorescence development in rice” to the International Journal of Molecular Sciences (IJMS). They previously characterized a mutant, apical panicle abortion1331 (apa1331), showing the abortion in apical spikelets. In this study, they studies the same mutant and quantified five different hormones and also performed RNA-seq analysis, and reported some associated genes. In GO analysis, they found significant terms associated with antioxidant activity, peroxidase activity and oxidoreductase activity. In KEGG analysis, they found downregulation of the biological processes including cell wall polysaccharides and fatty acids derivatives in the mutant. Based on expression values, they predicted some candidate genes and suggested that they can be potential targets of inflorescence development and might play an important role in panicle abortion. This study provides comprehensive information regarding rice spikelet abortion and inflorescence development. The manuscript is written well and the results are presented comprehensively. Anyhow, it needs some important revisions and adding relevant information before consideration in IJMS.

Figure 1; please arrange asterisks properly.

Figure 2; It is about basic transcript analyses. Authors can move it to a supplementary file.

L189; Cell division>cell division

Figure 3; please check the figure legend and revise it.

Figure 4; Legend,,, KEGG

The discussion section is weak. Please discuss obtained results relevant to previously published literature and your findings. Moreover, add some concluding remarks, some remaining work gaps, and possible future directions which can be drawn from this study.

Please use abbreviations properly. If a term is already abbreviated when it appeared first, then don’t abbreviate it again and again. The abstract, keywords and references are independent parts of the manuscript. In other sections, abbreviate considering them the whole manuscript.

The RNA-seq data was submitted to any repository? Please mention the accession numbers in the methodology section.

The authors used p<0.05, and FDR<0.05 for DEGs screening and GO and KEGG analyses? Please describe it in the methodology clearly. Moreover, providing information on p-values, and FDR for each gene/DEGs in an excel file containing FC values. Moreover, rename the excel tabs according to the data. (KEGG analysis, GO analysis, DEGs list, etc.). Moreover, you can put all the data in excel sheets into a single excel sheet with additional tabs. 

Author Response

Reviewer 1:

Authors submitted a manuscript entitled “Phytohormones and Transcriptome revealed the dynamics involved in spikelet abortion and inflorescence development in rice” to the International Journal of Molecular Sciences (IJMS). They previously characterized a mutant, apical panicle abortion1331 (apa1331), showing the abortion in apical spikelets. In this study, they studies the same mutant and quantified five different hormones and also performed RNA-seq analysis, and reported some associated genes. In GO analysis, they found significant terms associated with antioxidant activity, peroxidase activity and oxidoreductase activity. In KEGG analysis, they found downregulation of the biological processes including cell wall polysaccharides and fatty acids derivatives in the mutant. Based on expression values, they predicted some candidate genes and suggested that they can be potential targets of inflorescence development and might play an important role in panicle abortion. This study provides comprehensive information regarding rice spikelet abortion and inflorescence development. The manuscript is written well and the results are presented comprehensively. Anyhow, it needs some important revisions and adding relevant information before consideration in IJMS

We oblige the anonymous reviewer for his time and effort, as the revised version of the manuscript has been substantially improved with your suggestions. Our point-to-point responses to the reviewer’s comments are given below.

To improve the language and Grammar of the manuscript, we have carefully proofread and revised the manuscript with the help of all authors. The sentence structure has been thoroughly checked and improved in the revised version. Changes have been highlighted in red in the revised version.

Q: Figure 1; Please arrange asterisks properly.

Response: Thanks for the suggestion, we have carefully revised the asterisks in figure 1.

Q: Figure 2; It is about basic transcript analyses. Authors can move it to a supplementary file.

Response: Thanks for the suggestion, we have focused on your comment to move it to the supplementary figure but as it contains cluster analysis which is important and presented as the main results. With the advice of our senior authors, we decided to keep it as the main figure. If you still advise us to move to supplementary, please let us know. we can do further revisions.

Q: L189; Cell division>cell division

Response: We have carefully checked the paragraph and the specified point has been corrected.

Q: Figure 3; please check the figure legend and revise it.

Response: The figure legend has been revised. We corrected the mistake and changed B to C in the legend. The figure legend has been separated from the manuscript text.

Q: Figure 4; Legend,,, KEGG

Response: We have rectified the abbreviation now; it has been changed to KEGG.

The discussion section is weak. Please discuss obtained results relevant to previously published literature and your findings. Moreover, add some concluding remarks, some remaining work gaps, and possible future directions which can be drawn from this study.

Response: Please check the revised version, we have revised the discussion according to your suggestions.

Please use abbreviations properly. If a term is already abbreviated when it appeared first, then don’t abbreviate it again and again. The abstract, keywords and references are independent parts of the manuscript. In other sections, abbreviate considering them the whole manuscript.

Response: We have read the manuscript thoroughly and all the abbreviations have now been checked carefully and corrected accordingly.

The RNA-seq data was submitted to any repository? Please mention the accession numbers in the methodology section.

Response: Thanks for the suggestion, we have now mentioned the raw data accession numbers in the methodology section.

The authors used p<0.05, and FDR<0.05 for DEGs screening and GO and KEGG analyses? Please describe it in the methodology clearly. Moreover, providing information on p-values, and FDR for each gene/DEGs in an excel file containing FC values. Moreover, rename the excel tabs according to the data. (KEGG analysis, GO analysis, DEGs list, etc.). Moreover, you can put all the data in excel sheets into a single excel sheet with additional tabs.

Response: Thanks for the nice suggestion, we have already given DEGs filtration criteria in section 4.5.  We have provided FC values and P values against each of the DEGs. We also combined all the data into a single file and renamed as suggested.

Reviewer 2 Report

Comments to authors

 MS#ijms-1790915

 The author investigated the molecular pathways linked to spikelet abortion and inflorescence growth in rice using a transcriptome method in this regular work named "Phytohormones and Transcriptome revealed the dynamics involved in these processes in rice". The author also discussed how rice spikelet abortion and inflorescence growth are controlled by phytohormones. After reading the entire MS, I concluded that the current format may have been improved upon in terms of the data presentation style, the interpretation of the research gap, and the discussion of this study. However, I have some concerns about the current draft, such as:

Major concerns:

1.       The level of text writing is unsatisfactory; please take assistance of a native colleague/ speaker with experience in a related research subject try to improve it. Please make an effort to maintain consistency when describing plants, research gaps, prior studies, and the study's shot outline in the introduction section.

2.       Please provide a phenotype of WT and apa1331 so that the reader can understand the findings associated with spikelet abortion and inflorescence development. Moreover, the author did not show any comparison of agronomic traits between WT and apa 1331 (rate panicle spikelet degeneration, primary branches, lower primary spikelet, floral organs etc), and how inflorescence development influenced by these traits.

3.       Discussion is one of the important parts of an article. Section 3 is short, it should increase. A good discussion contains-i) Principles and relationship which can be supported by the results; ii) emphasis on results and conclusions that agree and disagree with other work(s); iii) Avoiding the repetition of the results in discussion sections; iv) theoretical implications or short summary/findings/suggestions of each section(s) where applicable.

4.       It would be fine if the author would maintain a consequence in data presentation, such as morpho-physiological>microscopical (SEM image….etc)>biochemical (phytohormone analysis, antioxidants, oxidative stress indicators) and molecular (transcriptomic or other molecular if have).

5.       In case Figure 3 and 4: author should clarify the significance of Fig. 3 and 4. As the author already mentioned the functional analysis of DEGs through clustering. I think, gene ontology (GO) should be considered only the three major aspects (molecular, biological and cellular), not based on hierarchical clustering. However, neither the antioxidant nor redox-state associated enzymatic nativities have been shown despite the fact that the author provided the processes connected to redox and antioxidants in figure 3. These all issues should be noticed.

Minor concerns:

1.       Less attention has been given for inflorescence development

2.       Abstract is bit lengthy. Try to concise it.

3.       L24-25: revise it with proper phrases.

4.       Plural form should be avoided in key word section

5.       Sort section should be assembled. Total three or four section is enough for introduction section.

6.       L120-142, 4 short sections are not looking good. These can be combined to total two.

7.       Fig. 1, L 146, found two dot (.), one dot should be deleted.

8.       In Fig 2A, front size line numbers are so small, so front size should be increased.  

9.       In figure 3, the arrangement of figure number (A, B, C, D), and position of dot (.) and comma (,) should be revised precisely.

10.    Figure caption of Fig.6 is incomplete. Please describe in detail. What does it mean the number of 1 to 12?

11.    Please avid the insertion of figure numbers in discussion, as these are already inserted in result section.

12.    L367, description of section 4.2.2 is too short. Please describe in detail with proper citation.

13.    Conclusion will not like abstract. It should be summarized with evidences of study and final suggestion(s). Please revise it.

14.    Very old references should be replaced by updated ones

Author Response

Reviewer 2:

 MS#ijms-1790915

 The author investigated the molecular pathways linked to spikelet abortion and inflorescence growth in rice using a transcriptome method in this regular work named "Phytohormones and Transcriptome revealed the dynamics involved in these processes in rice". The author also discussed how rice spikelet abortion and inflorescence growth are controlled by phytohormones. After reading the entire MS, I concluded that the current format may have been improved upon in terms of the data presentation style, the interpretation of the research gap, and the discussion of this study. However, I have some concerns about the current draft, such as:

Major concerns:

  1. The level of text writing is unsatisfactory; please take assistance of a native colleague/ speaker with experience in a related research subject try to improve it. Please make an effort to maintain consistency when describing plants, research gaps, prior studies, and the study's shot outline in the introduction section.

Response: To improve the language and Grammar of the manuscript, we have carefully proofread and revised the manuscript with the help of all authors. The sentence structure has been thoroughly checked and improved in the revised version. Changes have been highlighted in red in the revised version.

  1. Please provide a phenotype of WT and apa1331so that the reader can understand the findings associated with spikelet abortion and inflorescence development. Moreover, the author did not show any comparison of agronomic traits between WT and apa1331 (rate panicle spikelet degeneration, primary branches, lower primary spikelet, floral organs etc), and how inflorescence development influenced by these traits.

Response: Thanks for the suggestion, however, in our previous studies we have provided all suggested details of phenotypic and agronomic data. The current study is a further extension of our previous study (Ali, A., Wu, T., Zhang, H., Xu, P., Zafar, S. A., Liao, Y., ... & Wu, X. (2022). A putative SUBTILISIN-LIKE SERINE PROTEASE 1 (SUBSrP1) regulates anther cuticle biosynthesis and panicle development in rice. Journal of Advanced Research). We have already cited this article in our current study. To avoid repetition, with the advice of our senior authors we decided not to provide the phenotypic and agronomic data of WT and apa1331.

  1. Discussion is one of the important parts of an article. Section 3 is short, it should increase. A good discussion contains-i) Principles and relationship which can be supported by the results; ii) emphasis on results and conclusions that agree and disagree with other work(s); iii) Avoiding the repetition of the results in discussion sections; iv) theoretical implications or short summary/findings/suggestions of each section(s) where applicable.

Response: Thanks for the suggestion, your enlisted keypoints helped us to improve the discussion. Please see the revised version.

  1. It would be fine if the author would maintain a consequence in data presentation, such as morpho-physiological>microscopical (SEM image….etc)>biochemical (phytohormone analysis, antioxidants, oxidative stress indicators) and molecular (transcriptomic or other molecular if have).

Response: All physiological and microscopy data has already been published in our previous study Ali et al 2022 (Ali, A., Wu, T., Zhang, H., Xu, P., Zafar, S. A., Liao, Y., ... & Wu, X. (2022). A putative SUBTILISIN-LIKE SERINE PROTEASE 1 (SUBSrP1) regulates anther cuticle biosynthesis and panicle development in rice. Journal of Advanced Research). In this study, we aimed to dissect the differential changes of phytohormones and elucidated the physiological mechanism involved in panicle degeneration. Additionally, we predicted the differential targets involved in panicle development or abortion.

  1. In case Figure 3 and 4: author should clarify the significance of Fig. 3 and 4. As the author already mentioned the functional analysis of DEGs through clustering. I think, gene ontology (GO) should be considered only the three major aspects (molecular, biological and cellular), not based on hierarchical clustering. However, neither the antioxidant nor redox-state associated enzymatic nativities have been shown despite the fact that the author provided the processes connected to redox and antioxidants in figure 3. These all issues should be noticed.

Response: Thanks for the suggestion, in the revised version we have clarified the significance of using hierarchical clustering and Parametric Gene Set Enrichment (PGSEA) at the beginning of figure 3 and 4 results, respectively.

Figure 3: We believe Hierarchical clustering is the main strength of this study and we prefer its use on the following basis over simple GO. Hierarchical clustering can estimate the distance among overlapped genes with associated terms. Each gene can belong to multiple clusters rather than only one cluster and can be used to find a relationship among GO terms based on the distance among the overlapped genes. If the gene annotations are not complete then hierarchical clustering provides more robustness and accuracy with the use of similarity measures through R packages. Liu, M., Thomas, P.D. GO functional similarity clustering depends on a similarity measure, clustering method, and annotation completeness. BMC Bioinformatics 20, 155 (2019). https://doi.org/10.1186/s12859-019-2752-2

Figure 4: PGSEA is used to determine the list of genes that were downregulated in specific treatment and compare them to other datasets. Please note that PGSEA is independent of selected DEGs, it uses FC values of all genes in all samples. That’s why the processes of redox-state and antioxidants were not specifically included as these mentioned pathways belong to individual genes based on the FC values rather than common gene ontology.

Minor concerns:

  1. Less attention has been given for inflorescence development

Response: We have updated the text and modified it with the mentioned concern.

  1. Abstract is bit lengthy. Try to concise it.

Response: Thanks for the suggestion, we have tried our best to reduce redundant ideas from the abstract. The revised version of the abstract is sounding more precise now.

  1. L24-25: revise it with proper phrases.

 Response: We are sorry for mentioned mistake; we have revised the mentioned sentence.

  1. Plural form should be avoided in key word section

Response: We have changed all the keywords to their singular form in the keyword section.

  1. Sort section should be assembled. Total three or four section is enough for introduction section.

Response: We have merged five different paragraphs into three in the introduction section.

  1. L120-142, 4 short sections are not looking good. These can be combined to total two.

Response: We have merged shorts section into two.

  1. 1, L 146, found two dot (.), one dot should be deleted.

Response: We have removed corrected the mistake.

  1. In Fig 2A, front size line numbers are so small, so front size should be increased.  

Response: We have corrected the visibility issue in the font size of Fig.2A

  1. In figure 3, the arrangement of figure number (A, B, C, D), and position of dot (.) and comma (,) should be revised precisely.

Response: We have revised the figure legends of Fig. 3 carefully and corrected the punctuation mistakes.

  1. Figure caption of Fig.6 is incomplete. Please describe in detail. What does it mean the number of 1 to 12?

Response: Sorry for the mistake, we have revised the figure legends and 1-12 represents the number of chromosomes of rice.

  1. Please avid the insertion of figure numbers in discussion, as these are already inserted in result section.

Response: We have removed the figure number citations from the discussion part.

  1. L367, description of section 4.2.2 is too short. Please describe in detail with proper citation.

Response: Thanks for suggestion. We have prepared the samples of GA as mentioned in section 4.2.1 and as per suggestion we added a citation in section 4.2.2.

  1. Conclusion will not like abstract. It should be summarized with evidences of study and final suggestion(s). Please revise it.

Response: Thanks for the suggestion, we have revised the conclusion part by adding the pieces of evidence of study and final suggestions.

  1. Very old references should be replaced by updated ones

Response: We understand that we have cited 2-3 old references, however, with the discussion with our senior authors we aimed to yet retain these references due to the followings.

There are no currently new references working specifically with the working theories of panicle degeneration, especially about resource limitation and self-organization theory. Although many studies have been reported in favor or against them. Still, these references belong to impactful and well-reputed journals, it would be an injustice to not cite the basic studies entailing the physiological basis of panicle degeneration. However, if you still advise us to remove them, please let us know we will remove them in further revisions.

Reviewer 3 Report

The manuscript presented for review concerns an interesting issue related to panicle degradation in rice, which has a direct impact on the yield of this important plant from an economic point of view. The results obtained are interesting, but I have some comments regarding the manuscript. Discussion is the weakest part of the manuscript, especially subsection 3.2, which lacked a deeper confrontation of the obtained results with the literature. This subsection seems to be the key in this work, so it needs to be improved, because in its present form it is a collection of quite obvious statements. Furthermore, in subsection 3.1, lines 303-304, requires the elaboration of the role of ABA in the mentioned process.  Moreover, there is newer literature on this issue, not from over 20 years ago. I also have a few minor comments regarding editing:

lines 109-111 and 120-122 are nearly identical, unnecessary repetition

line 204 the caption of figure 3 lacks an explanation of part C of the figure, twice is the description of part A

line 204 the text of the manuscript is linked to the figure caption, it need to be separated

lines 222-226 are part of the discussion, not a description of the results

lines 266-271 are formatted as a table caption, not the manuscript text, it need to be changed

lines 277-280 the primer sequences should be in the methods, not in the description of the results

In addition, the references should be unified, and the abbreviation titles of journals should be used.

Author Response

Reviewer 3:

The manuscript presented for review concerns an interesting issue related to panicle degradation in rice, which has a direct impact on the yield of this important plant from an economic point of view. The results obtained are interesting, but I have some comments regarding the manuscript. Discussion is the weakest part of the manuscript, especially subsection 3.2, which lacked a deeper confrontation of the obtained results with the literature. This subsection seems to be the key in this work, so it needs to be improved, because in its present form it is a collection of quite obvious statements. Furthermore, in subsection 3.1, lines 303-304, requires the elaboration of the role of ABA in the mentioned process.  Moreover, there is newer literature on this issue, not from over 20 years ago. I also have a few minor comments regarding editing:

We thank the anonymous reviewer for his time and effort, as the revised version of the manuscript has been substantially improved with your suggestions. Our point-to-point responses to the reviewer’s comments are given below. To improve the language and Grammar of the manuscript, we have carefully proofread and revised the manuscript with the help of all authors. The sentence structure has been thoroughly checked and improved in the revised version. Changes have been highlighted in red in the revised version.

The subsection 3.2 in the discussion part has been thoroughly read and the shortcoming of connection between results and previous work has been fulfilled to our best. Similarly, the lines 303-304 of the subsection 3.1 has been revised and elaborated for the role of ABA.

Following are the minor comments.

Q: lines 109-111 and 120-122 are nearly identical, unnecessary repetition

Response: Thanks for the careful review, we have deleted the redundant information given in 120-122 (Results).

Q: line 204 the caption of figure 3 lacks an explanation of part C of the figure, twice is the description of part A

Response: We have rectified the text. The B has been replaced by C and the commas have been inserted after each part (A, B and C). 

Q: line 204 the text of the manuscript is linked to the figure caption, it need to be separated

Response: Thanks for pointing, the mistake has been identified and rectified accordingly.

Q: lines 222-226 are part of the discussion, not a description of the results

Response: The lines 222-226 have been checked and have been deleted from the results and have been added to the discussion part.

Q: lines 266-271 are formatted as a table caption, not the manuscript text, it need to be changed

Response: The lines 266-271 have now been changed from table caption to the manuscript text.

Q: lines 277-280 the primer sequences should be in the methods, not in the description of the results

Response: We have removed the primer sequences from the lines 279-280 and have been added them in the methods part 4.7.

Q: In addition, the references should be unified, and the abbreviation titles of journals should be used.

Response: We have unified the abbreviations and made changes in all references to set them according the journal guidelines.

Round 2

Reviewer 1 Report

Authors have addressed most of the comments. I’ve few more questions to ask before considering this manuscript for publication.

In 4.5. section, authors have mentioned FDR below 0.1. They used 0.05 or 0.1? Moreover, in supplementary excel file, authors only mention P-adj (Adjusted P-value), not p-value. Please provide p-value and adj p-value together in excel file. Statistical analysis should be very clear; however, it will be confusing for readers.

Generally, it is not recommended attempting WGCNA on a data set consisting of fewer than 12 samples. Correlations on fewer than 12 samples mostly be too noisy for the network to be biologically meaningful. In most of the cases, it is recommended that one should have at least 20 samples; as with any analysis methods, more samples typically lead to more vigorous and refined results. In this study there were two different samples (6 in total). How can authors assure the reliability of data analysis?

Author Response

Reviewer 1:

In 4.5. section, authors have mentioned FDR below 0.1. They used 0.05 or 0.1? Moreover, in supplementary excel file, authors only mention P-adj (Adjusted P-value), not p-value. Please provide p-value and adj p-value together in excel file. Statistical analysis should be very clear; however, it will be confusing for readers.

Response: Thanks for asking, We have used iDEP .0951 reported by Ge, S.X.; Son, E.W.; Yao, R. iDEP: an integrated web application for differential expression and pathway analysis of RNA-Seq data. BMC bioinformatics 2018, 19, 1-24. We can set P-values for DEGs in the background and it automatically adjusts the P-values using default parameters and finally, we only can download adjusted P-values. We are sorry, the list we retired from the software does not show p-values.

We have changed the terms “p-values” to “adjusted p-values” in the text and also mentioned (The p-values were adjusted for multiple testing using the default method integrated with iDEP 0.951) in the revised text.

Regarding FDR, we used default FDR below 0.1 which is correctly mentioned in section 4.5 and throughout the manuscript.

Generally, it is not recommended attempting WGCNA on a data set consisting of fewer than 12 samples. Correlations on fewer than 12 samples mostly be too noisy for the network to be biologically meaningful. In most of the cases, it is recommended that one should have at least 20 samples; as with any analysis methods, more samples typically lead to more vigorous and refined results. In this study there were two different samples (6 in total). How can authors assure the reliability of data analysis?

Response: Thanks for pointing out an important concern, we agree attempting WGCNA analysis on fewer than 15 samples may result in a noisy network. Although, few researchers claim that the sample size may not be a decisive factor, but comentropy dose, especially when you are using SD/mean rather than the expression values. As most of the researchers are not in favor to do WGCNA analysis if samples are less than 15, we have removed these results and discussion related to this from all parts of the manuscript.

Reviewer 2 Report

Comments to the authors

 MS#ijms-1790915-R1

 The author revised most of points raised by the reviewer. However, some minor points still exists. These should be addressed before it further processing, such as:

 1. The title seems a little ambiguous. The term "analysis" can be added to complete and encompass the entire story. "Phytohormones and Transcriptome analyses revealed the dynamics involved in spikelet abortion and inflorescence development in rice" would be the new title.

  2. P3, L124-25: It is preferable to eliminate the tiny space between the two lines.

  3. In place of the figure area, the author of Figure 1's caption should include the meaning of different asterisks

  4. Please elaborate on the caption of Fig. 3 in detail with bioinformatics tools.

Author Response

Reviewer 2:

  MS#ijms-1790915-R1

 The author revised most of points raised by the reviewer. However, some minor points still exists. These should be addressed before it further processing, such as:

  1. The title seems a little ambiguous. The term "analysis" can be added to complete and encompass the entire story. "Phytohormones and Transcriptome analyses revealed the dynamics involved in spikelet abortion and inflorescence development in rice" would be the new title.

Response: Thanks for the suggestion, we have revised the title as suggested.

  1. P3, L124-25: It is preferable to eliminate the tiny space between the two lines. Response: Sorry for the mistake, we have removed the space as suggested.

  1. In place of the figure area, the author of Figure 1's caption should include the meaning of different asterisks

Response: Thanks for the suggestion, we have added the caption in Figure 1’s legend and removed it from within the figure.

  1. Please elaborate on the caption of Fig. 3 in detail with bioinformatics tools.

Response: We have revised the figure 3 legend and added details of bioinformatic tool parameters.